# Monitoring and Assessment of the Oasis Ecological Resilience Improved by Rational Water Dispatching Using Multiple Remote Sensing Data: A Case Study of the Heihe River Basin, Silk Road

**Jiaxin Du** [1,2], **Bihong Fu** [1,*], **Qiang Guo** [1] **and Pilong Shi** [1]

1   Aerospace Information Research Institute, Chinese Academy of Sciences, Beijing 100094, China; dujx@radi.ac.cn (J.D.); guoqiang@radi.ac.cn (Q.G.); shipl@radi.ac.cn (P.S.)
2   University of Chinese Academy of Sciences, Beijing 100049, China
*   Correspondence: fubh@radi.ac.cn; Tel.: +86-10-8217-8096

**Abstract:** The suboptimal management and utilization of water resources from the "Asian water towers" contributed to serious ecological crises in river basins along the arid Silk Road, such as the Aral Sea and the Heihe River in the 20th century. To improve the ecological resilience of the Ejina Oasis in the Heihe River downstream basin, the Chinese government implemented the 'Ecological water dispatching project' in 2000. However, it is still unclear what the optimal rational water allocation is for the sustainable development of economic, social, and ecological environments (so called "triple bottom line") in these inland river basins. This study presents a decision-tree-based methodology for ecological monitoring and restoration strategies for Silk Road's oasis eco-system. Using Landsat TM/OLI data as well as meteorological, hydrological, and water utilization data, we show that ~69% of the originally degraded land has been restored since 2000. Previously dry tail-end lakes in the Heihe River downstream basin have been rejuvenated, and the precipitation has also significantly improved ($\rho = 0.047$). We propose that the downstream water allocation should be no more than $\sim 11 \times 10^8$ m$^3$ and that the optimal ratio between downstream and midstream allocation is 0.4–1.7. This study provides an excellent example for ecological monitoring and assessment in the optimization of strategies for the restoration of Silk Road's oasis eco-system.

**Keywords:** oasis ecosystem resilience; water allocation; desertification assessment; multi-source remote sensing; Silk Road

## 1. Introduction

Water management [1] is critical for improving ecological resilience [2] in arid oases along the Silk Road, a historically important trade route of central Asia [3,4]. The term "Asian water towers" (AWT) encompasses the mountain river systems in Asia that drain the Tibetan Plateau fed by snow and melting glaciers. The AWT supplies a substantial part of the water demands of nature and people in Asia [1,5]. In the 20th century, the vulnerability of AWT [5] to the mismanagement of water led to environmental crises [2] in downstream oases such as the Aral Sea basin and the Heihe River basin. In central Asia, the Aral Sea water crisis reflects the overconsumption of water for agricultural purposes and energy production since the 1960s [6]. Similarly, the Heihe River downstream basin (HRDB) also faces serious desertification and salinization due to the excessive use of water for the expansion of agriculture and population growth within the midstream along the Hexi Corridor since the 1950s [7].

The Heihe River basin, covering 130,000 km$^2$, originated from the Qilian Mountain range in the northeastern Tibetan Plateau and is the second largest inland river basin in China [8] (Figure 1).

As an important part of AWT, the Qilian Mountain contributes to 80.2% of the total upstream flow of the Heihe River [8,9]. In the northern piedmont of Qilian Mountain, the midstream, socio-economic development is estimated to have diverted 86% of the total water of the Heihe River basin [10] since the 1950s, with serious ecological problems for the HRDB [8]. For example, Advanced Very High-Resolution Radiometer (AVHRR) time series data indicated a sharp vegetation decrease of 3240 km$^2$ in the HRDB from the 1980s to 1990s [11]. The tail-end lake of the Heihe River (East Juyan lake) dried up in the early 1990s [12]. Spatio-temporal changes in the Normalized Difference Vegetation Index (NDVI) indicated that the average groundwater table-level fell by about 3.5 m from 1989 to 2000 in the Ejina oasis of the HRDB [13].

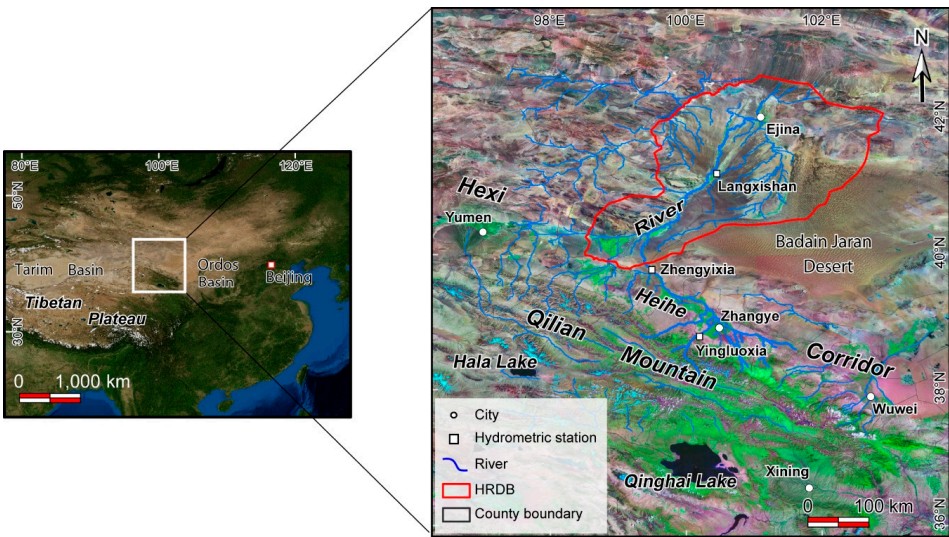

**Figure 1.** Satellite images showing the location of the study area. HRDB—Heihe River downstream basin.

To improve the ecological resilience of the Ejina oasis, the Chinese government resolved to invest 2.35 billion RMB Yuan to implement the 'Ecological Water Dispatching Project' in 2000 [8], allowing the midstream dispatch of $9.5 \times 10^8$ m$^3$ of water to the HRDB each year [14]. This water dispatching has seen the substantial recovery of the *Populus euphratica*, desert shrubberies, and meadow grasses in the HRDB [15], as well as a rise of the groundwater table level by 0.5–1.6 m [16,17]. Meanwhile, there has been an intense salinization of the soil in the HRDB from slightly saline to moderately and intensely saline [18]. In addition, it has been argued that the over-diversion of water has resulted in an imbalance in the midstream, resulting in desertification along the whole river basin [8]. Consequently, a current estimate of the optimal water allocation for sustainable development along the Heihe River basin remains pressing and is as-of-yet undetermined for the "triple bottom line" [19] (i.e., economic, social, and ecological development) in the oases along the Silk Road.

Remote sensing technology has proven to be a cost-effective approach for the assessment of desertification [20–23] and ecological restoration [24,25]. Examples include: (1) the application of Spectral Mixture Analysis (SMA) to Landsat images for land degradation in Argentina [26]; (2) the use of Landsat, AVHRR, and Indian Remote Sensing Satellite (IRS)-1C data to assess the impacts of climate and humans on rangeland desertification, which were distinguished based on NDVI in Syria [27]; (3) the use of AVHRR for the calculation of Rain Use Efficiency (RUE), which suggested that constant RUE was the preliminary evidence in assessing land degradation [28]; (4) the use of 250-m MODIS NDVI and GIS, which indicated that an area of approximately 161,000 ha suffered desertification at different levels in Central Asia, associated with Aral Sea Basin shrinkage due to irrational over-irrigation [29]; and (5) the use of multi-Landsat images from 1975 to 2010 to assess human impacts on 47,833 km$^2$ of aeolian deserted land in northwest China [22].

Because desertification not only causes serious ecological feedback but also leads to major social problems [30], remote sensing monitoring studies such as those listed above can have important social policy ramifications. While remote sensing data have been previously employed to monitor the dynamics of vegetation, water, soil, and evapotranspiration in the Heihe River basin [11,12,31–34], there is still a lack of comprehensive monitoring and assessment of spatio-temporal desertification status and ecological dynamics in the HRDB.

In order to determine the impact of water dispatching on the ecological environment and climate changes in HRDB, this study employed the multi-temporal Landsat satellite data and decision tree to assess the desertification dynamics in HRDB before and after water dispatching (from 1995 to 2015). Accordingly, a rational water dispatching threshold is proposed for maintaining and improving ecological resilience in the oases along the Silk Road.

## 2. Materials and Methods

### 2.1. Study Area

The HRDB (between 98°30′–102°53′ E and 39°47′–42°37′ N) is located to the north of the Zhengyixia hydrological station, west of the Badain Jaran Desert and in the northern front of Qilian Mountain, covering an area of approximately 66,574.9 km$^2$ (Figure 1). In the HRDB, the Heihe River is approximately 333 km long, with sand and gravel desert on both its banks [35]. The climate in the HRDB is arid with a mean annual precipitation of only 34–47 mm and an average annual temperature of nearly 10 °C [12]. The Ejina oasis is an isolated oasis located at the north end of the HRDB, acting as a barrier to the desert transition [34]. Vegetation is dominated by *Populus diversifolia* and *Sacsaoul*, both of which are tolerant to aridity and therefore play a key role in impeding desertification.

### 2.2. Processing of Landsat Data for Desertification Assessment

Landsat-5 (TM) and Landsat-8 (OLI) level-1 terrain corrected (L1T) images with a spatial resolution of 30 m acquired in 1995, 2000, 2005, 2010, and 2015 were downloaded from the Geospatial Data Cloud (GDC) (http://www.gscloud.cn) and the United States Geological Survey (USGS) (http://glovis.usgs.gov). Four Landsat scenes cover the study area with a path/row of 133/31, 133/32, 134/31, and 134/32 (Table 1). The cloud-free satellite data obtained from July to September were chosen because this is when the vegetation generally reaches the maximum coverage. The radiometric, geometric, and atmospheric correction and clipping processing of these images for each period were carried out in the Environment for Visualizing Images (ENVI) software.

**Table 1.** Basic information of Landsat imagery.

| Type | Dates | Path/Row | Spatial Resolution | Source |
|---|---|---|---|---|
| Landsat-5 TM | 19950821 | 133/31 133/32 | 30 m | USGS |
| | 19950913 | 134/31 134/32 | 30 m | USGS |
| | 20000708 | 134/31 134/32 | 30 m | GDC |
| | 20000818 | 133/31 133/32 | 30 m | GDC |
| | 20050816 | 133/31 133/32 | 30 m | USGS |
| | 20050908 | 134/31 134/32 | 30 m | USGS |
| | 20100814 | 133/31 133/32 | 30 m | GDC |
| | 20100821 | 134/31 134/32 | 30 m | GDC |
| Landsat-8 OLI | 20150718 | 134/31 134/32 | 30 m | USGS |
| | 20150828 | 133/31 133/32 | 30 m | USGS |

### 2.3. Indicative Index and Decision Tree for Desertification Assessment

The Food and Agriculture Organization (FAO) and the United Nations Environment Programme (UNEP) developed a methodology with 22 indicators to assess desertification [21,36]. Water, vegetation coverage, and soil status are the key factors reflecting the desertification conditions in arid and semi-arid regions [37,38]. The NDVI, SMA, and RUE are indices frequently used to quantify the dynamics of vegetation, soil, and rainfall use for desertification assessment [20,21,28,39,40]. However, the NDVI is seriously affected by the rainfall and background value of the soil [40,41]. The use of SMA is a precise approach for quantitative measurements of vegetation and soil land but shows limitation in areas covered with sparse vegetation [38]. The accuracy of RUE strongly depends on the NDVI, which is suitable for areas with annual rainfall of more than 100 mm [20,28]. Therefore, in this study, three indicators—(1) the Modified Normalized Difference Water Index (MNDWI) [42], (2) the Modified Soil Adjusted Vegetation Index (MSAVI) [41], and (3) the Bare Soil Index (BSI) [43]—were combined to comprehensively reflect water, vegetation, and soil status for the assessment of the desertification dynamics in the HRDB, where vegetation is sparse and the annual rainfall is less than 50 mm/a.

The MNDWI is applied to extract water in arid and semi-arid regions, particularly in the HRDB, where runoff has a greater impact than rainfall [8]. The MNDWI provides an effective means for analyzing areas that are sensitive to changes in ground water, such as lakes, rivers, and reservoirs, which can be calculated as the following [42]:

$$\text{MNDWI} = \frac{\rho_G - \rho_{NIR}}{\rho_G + \rho_{NIR}} \tag{1}$$

To minimize the soil background's impacts and increase the vegetation sensitivity, the MSAVI was selected to monitor the vegetation dynamics and calculated as follows [41]:

$$\text{MSAVI} = \left(2\rho_{NIR} + 1 - \sqrt{(2\rho_{NIR} + 1)^2 - 8(\rho_{NIR} - \rho_R)}\right)/2 \tag{2}$$

It is worth noting that the value of the vegetation index may not be reliable if the vegetation is sparse [43]. Therefore, the BSI was proposed to improve the evaluation of the vegetation status, as the lower the vegetation value, the higher the BSI value [43]. The BSI is obtained using [43]:

$$\text{BSI} = \frac{(\rho_{SWIR} + \rho_R) - (\rho_{NIR} + \rho_B)}{(\rho_{SWIR} + \rho_R) + (\rho_{NIR} + \rho_B)} \tag{3}$$

In Equations (1)–(3), $\rho_B$, $\rho_G$, $\rho_R$, $\rho_{NIR}$, and $\rho_{SWIR}$ represent spectral bands 1, 2, 3, 4, and 5 for Landsat-5 TM imagery and bands 2, 3, 4, 5, and 6 for Landsat-8 OLI imagery, respectively.

Decision trees [44] have been used in a number of studies to assess desertification [20,21,45,46]. Generally, the degree of desertification is classified into five categories: Non-degraded, Low, Moderate, High, and Severe desertification [47,48]. To obtain the thresholds for each of these five grades, each index was firstly extracted according to spectral calculation Equations (1)–(3) for processed the Landsat imagery of each period in ENVI. Then, a total of 1500 training samples for each index, including 300 training samples of each grade, were selected, referring to GlobeLand30 data (http://www.globallandcover.com/GLC30Download/) and ground features according to Google Maps. Among the processed Landsat images for five periods, a total of 1500 inspection points were selected for each period, including 300 random validation points of each grade. The decision tree rule for all the thresholds was obtained by using the ENVI see 5.0 model [20]. Figure 2 indicates the range of values for each index for each desertification degree. Finally, the decision tree was applied to classify the desertification of each period. The overall accuracy of the classification results was verified by using the inspection points of each period.

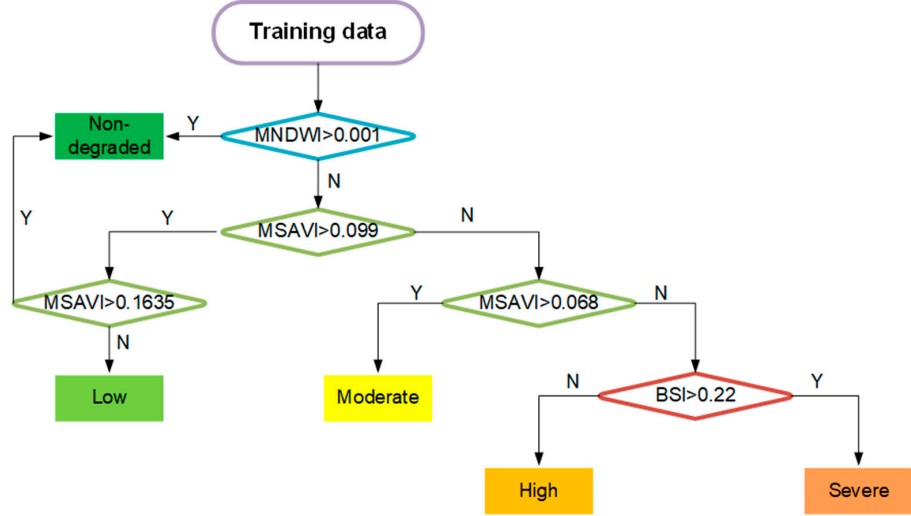

**Figure 2.** Decision tree for the assessment of the desertification situation. There are five grades of desertification degree, including Non-Degraded, Low, Moderate, High, and Severe desertification. The three indices are the MNDWI—Modified Normalized Difference Water Index, MSAVI—Modified Soil Adjusted Vegetation Index, and BSI—Basic Soil Index. The decision tree rule shows the range values of each index for determining grades of desertification.

### 2.4. Meteorological, Hydrological and Water Utilization Data

The meteorological data of the monthly average precipitation and temperature acquired from 12 meteorological stations in and around the Heihe River area from 2000 to 2015 were downloaded from the China Meteorological Administration (http://data.cma.cn/) for the spatial analysis of climate change in the HRDB. The TerraClimate monthly datasets [49] downloaded from the Climatology Lab (http://www.climatologylab.org/terraclimate.html/) were used to determine the spatial annual precipitation of Heihe River area from 2000 to 2015. Additionally, the hydrological data of the upstream, midstream, and downstream runoff of the Heihe River recorded at the Yingluoxia, Zhengyixia, and Langxinshan hydrological stations, respectively, were obtained from the Ministry of Water Resources the People's Republic of China (http://www.mwr.gov.cn/sj/). Water utilization data for industrial and domestic use were calculated using data from the Bulletin of Water Resources of Inner Mongolia Autonomous Region from 2000 to 2015, downloaded from the Water Resources Department of Inner Mongolia Autonomous Region (http://slt.nmg.gov.cn/).

### 2.5. Spatio-Temporal Analyses of Meteorological Data

Kriging interpolation is a well-established and important method for the spatial analyses of meteorological data [50]. The advantage of Kriging is that it couples with a linear combination of the weighted spatial structure changes and spatial autocorrelation [50,51]. Interpolation can be conducted in the Kriging spatial analyst module in the ArcGIS 10.2 software. The average annual temperature and precipitation data for the HRDB were calculated from the monthly climate meteorological dataset with interpolation using ordinary Kriging with Spherical semivariogram models applied to annual spatial temperature and precipitation.

Cross validation used to estimate the Mean Error (ME), Root-Mean-Square Error (RMSE), Mean Standardized Error (MSE), and Root-Mean-Square Standardized Error (RMSSE) of the Kriging interpolation results [50]. To improve the reliability of the Kriging interpolation results, the TerraClimate monthly datasets were calculated to obtain annual data in ArcGIS 10.2, which provided evidence of the spatial variations in precipitation over the period 2000–2015 [49]. The temporal trend of the changes in precipitation and temperature was tested using non-parametric Mann–Kendall tests [52,53].

The correlation of the precipitation and temperature data with the annual runoff was calculated to explore the impact of the runoff on the spatio-temporal variations in the climate of the HRDB.

### 2.6. Water Balance between Downstream and Midstream

To determine the water balance between the downstream and midstream of the Heihe River, water allocation thresholds for the downstream and midstream were calculated using the equations documented below [37,54,55].

The estimation of thresholds requires the average annual runoff water consumed by downstream and midstream, respectively, and takes the proportion of each type of water consumption from the total runoff into consideration.

$$W_T = W_D + W_M = p \times W_T + q \times W_T \tag{4}$$

The average annual runoff water demand is calculated by subtracting the total water consumption from precipitation.

$$W_{D/M} = P_{D/M} \times A - W \tag{5}$$

The total water consumption consists of the average annual evapotranspiration of ground and vegetation, industrial water use, and domestic water use.

$$W = ET \times a + W_{Ind} + W_{Dom} \tag{6}$$

The variables in the equations above are defined as follows: $W_T$ is the average annual total runoff, $W_D$ is the average annual runoff demand for downstream, $W_M$ is the average annual runoff demand for midstream, $P_D$ is the average annual precipitation of downstream, $P_M$ is the average annual precipitation of midstream, $W$ is the total water consumption, $ET$ is the average annual evapotranspiration value, $W_{Ind}$ is the average annual water demand for industrial use, $W_{Dom}$ is the average annual water demand for domestic use, $A$ is the total area of the midstream or downstream region, $a$ is the area of the oasis including vegetation coverage and lakes, $p$ is the percentage of $W_D$ in $W_T$, and $q$ is the percentage of $W_M$ in $W_T$.

The total runoff is the long-term average annual runoff calculated from the data of Yingluoxia. The water demand amounts of the midstream are based on data from previous studies [56,57]. The water needed from the downstream is designed as the total water consumption excluding the precipitation in the HRDB. The total water consumption of the HRDB consists of the $ET$ of the ecological and socio-economic water demand. The $ET$ values from 2000 to 2015 were collected from previous studies [12,31,55,58,59]. The socio-economic water consumption includes the water demand for industrial and domestic use but excludes the agricultural water demand because the total ET includes the water use of crops. The precipitation was obtained by combining the records from 12 meteorological stations. The minimum and maximum water demands in the midstream and downstream are calculated based on the model. The water allocation threshold is determined as the ratio of downstream and midstream (*p*/*q*). The verification was based on the true value of the runoff in midstream and downstream from 2000 to 2015.

## 3. Results

### 3.1. Spatio-Temporal Variations in HRDB Ecological Environment from 1990 to 2015

The spatio-temporal dynamics of desertification in the HRDB are presented in Figure 3a,b. In total, from 1995 to 2015, the statistical results show significant increases in the area Non-Degraded (+1523 km$^2$, +182.4%) and with Low (+816 km$^2$, +78.7%), Moderate (+3461 km$^2$, +135.7%), and High desertification (+20,319 km$^2$, +84.4%). By contrast, the area of Severe desertification clearly reduced (−26,119 km$^2$, −68.6%) (Tables 2 and 3).

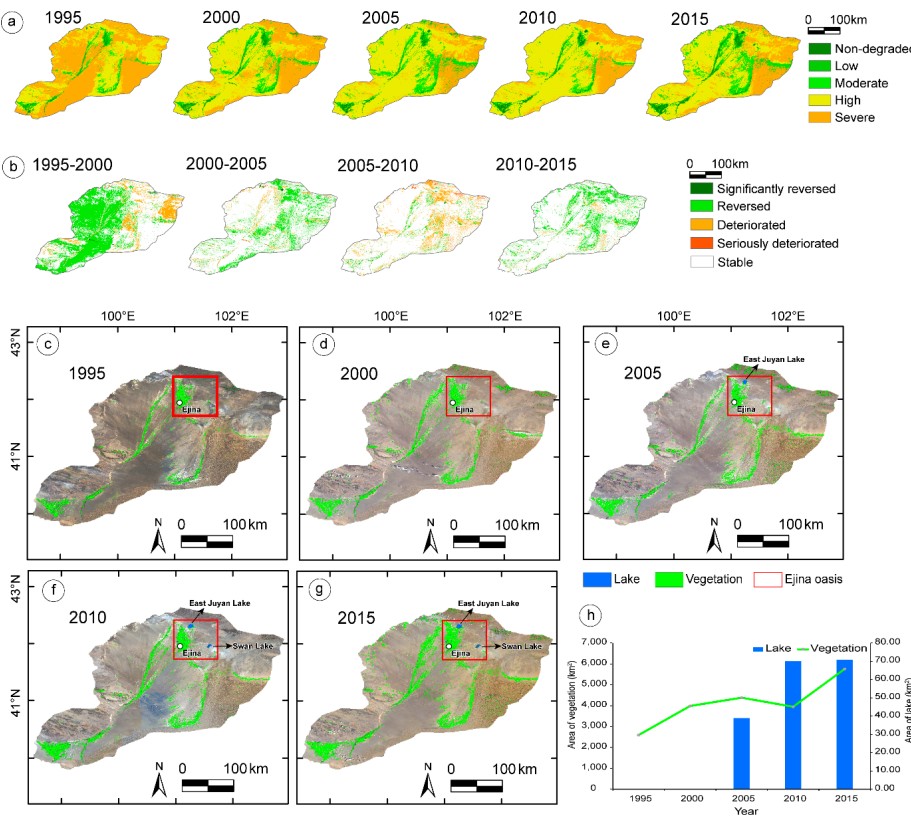

**Figure 3.** Desertification assessment results and spatio-temporal changes in vegetation and tail-end lakes in the HRDB from 1995 to 2015. (**a**) Desertification status of the HRDB from 1995 to 2015. (**b**) Spatial desertification dynamics of the HRDB in 1995–2000, 2000–2005, 2005–2010, and 2010–2015, respectively. Red rectangles present the dynamics of vegetation and tail-end lakes in the Ejina oasis during this period: (**c**) 1995; (**d**) 2000; (**e**) 2005; (**f**) 2010; (**g**) 2015. (**h**) Broken line and histogram graph showing the area variation of vegetation and lakes.

**Table 2.** The situation of desertification in the HRDB from 1995 to 2015.

| Desertification Degree | 1995 | | 2000 | | 2005 | | 2010 | | 2015 | |
|---|---|---|---|---|---|---|---|---|---|---|
| | Area (km²) | % of HRDB | Area (km²) | % of HRDB | Area (km²) | % of HRDB | Area (km²) | % of HRDB | Area (km²) | % of HRDB |
| Non-Degraded | 835.38 | 1.25 | 1465.59 | 2.20 | 1306.65 | 1.96 | 1678.70 | 2.52 | 2359.10 | 3.54 |
| Low | 1036.36 | 1.56 | 1203.17 | 1.81 | 1474.89 | 2.22 | 971.61 | 1.46 | 1851.86 | 2.78 |
| Moderate | 2550.70 | 3.83 | 4431.80 | 6.66 | 6421.46 | 9.65 | 2953.46 | 4.44 | 6011.93 | 9.03 |
| High | 24,063.91 | 36.15 | 38,222.18 | 57.41 | 44,446.15 | 66.76 | 45,156.20 | 67.83 | 44,383.14 | 66.67 |
| Severe | 38,088.54 | 57.21 | 21,252.16 | 31.92 | 12,925.75 | 19.42 | 15,814.93 | 23.76 | 11,968.87 | 17.98 |
| Overall Accuracy | 0.87 | | 0.92 | | 0.86 | | 0.85 | | 0.90 | |

**Table 3.** Dynamic changes of the land in the HRDB from 1995 to 2015.

| Desertification Degree | 1995–2000 | | 2000–2005 | | 2005–2010 | | 2010–2015 | | Total (1995–2015) | |
|---|---|---|---|---|---|---|---|---|---|---|
| | Area (km²) | % of 1995 | Area (km²) | % of 2000 | Area (km²) | % of 2005 | Area (km²) | % of 2010 | Area (km²) | % of 1995 |
| Non-Degraded | +630.21 | +75.44 | −158.94 | −10.84 | +372.05 | +28.47 | +680.40 | +40.53 | +1523.72 | +182.40 |
| Low | +166.81 | +16.10 | +271.72 | +22.58 | −503.28 | −34.12 | +880.25 | +90.60 | +815.50 | +78.69 |
| Moderate | +1881.10 | +73.75 | +1989.66 | +44.90 | −3468.01 | −54.01 | +3058.47 | +103.56 | +3461.23 | +135.70 |
| High | +14,158.27 | +58.84 | +6223.98 | +16.28 | +710.05 | +1.60 | −773.06 | −1.71 | +20,319.23 | +84.44 |
| Severe | −16,836.38 | −44.20 | −8326.42 | −39.18 | +2889.18 | +22.35 | −3846.06 | −24.32 | −26,119.67 | −68.58 |

The spatial dynamics of the vegetation coverage and lake water extent in the HRDB from 1995 to 2015 (Figure 3 and Table 4) were determined with the MSAVI and MNDWI. The extent of vegetation coverage was low before 2000. After water dispatching, it showed a significant upward trend (from

2591 to 5757 km$^2$) with an increase of 158 km$^2$/a. The results prove that the approach is a feasible way to monitor land cover through vegetation dynamics over large areas using Landsat images.

**Table 4.** Areas of and changes in vegetation in the HRDB and tail-end lakes of the Heihe River from 1995 to 2015.

| Year | Vegetation (MSAVI > 0.08) | | Lakes (MNDWI > 0.8) | |
|---|---|---|---|---|
| | Area (km$^2$) | Variation Area (km$^2$) | Area (km$^2$) | Variation Area (km$^2$) |
| 1995 | 2591.59 | 0 | 0 | 0 |
| 2000 | 3984.41 | 1392.82 | 0 | 0 |
| 2005 | 4380.84 | 396.43 | 38.93 | 38.93 |
| 2010 | 3947.32 | −433.52 | 70.13 | 31.20 |
| 2015 | 5757.40 | 1810.08 | 70.73 | 0.60 |
| Annual average increase | 158.29 | | 3.54 | |

It is noticeable that all the tail-end lakes were dry from 1995 to 2000. Figure 3 shows that the East Juyan Lake has been restored since 2005. To the east, another lake was also partially filled by 2010. In total, the areas of these lakes reached 70.73 km$^2$ in 2015. These results imply a significant restoration of the vegetation coverage and water resources in the HRDB following the implementation of ecological water dispatching in 2000.

In total, our ecological assessments indicate that approximately 26,119 km$^2$ of the desertified land was significantly restored due to water dispatching, accounting for 69% of the land that was degraded in the HRDB.

### 3.2. Spatio-Temporal Changes in the HRDB's Climate Since Water Dispatching was Implemented in 2000

The spatio-temporal changes in temperature and precipitation in the HRDB from 2000 to 2015 were analyzed based on meteorological data (Figure 4). The average annual temperature was 9.65 °C, with an increase of 0.06 °C/a, and the average annual precipitation was 37.6 mm, with an increase of 1.22 mm/a (Table S1). The non-parametric Mann–Kendall tests quantify the fluctuation of the climate change; the temperature had a consistent trend over this period, whereas the precipitation showed significant variability that was higher than average in the first two years after water dispatching and lower than average between 2003 and 2006, and it steadily increased from 2007 to 2015 (Figure S1).

As shown in Figure 4a, over the period of 2000–2005, the northeastern area became colder and drier, while the southwestern area became warmer and wetter. From 2005 to 2010, the northeastern area exhibited hotter and drier conditions than the southeastern area. Then, both temperature and precipitation significantly increased in the northeastern area from 2010 to 2015. The cross validation calculated the ME, RME, MSSE, and RMSSE of the Kriging interpolation of the precipitation and temperature (Tables S2 and S3). The validation of both the precipitation and temperature shows ME values of less than 1 except for the precipitation's ME in 2015. In addition, for both, the MSSE values are all approximately 0, with the RMSSE values approaching 1. These validation results imply that the overall interpolation error is small, despite the high RME values for precipitation. The RME values for temperature of ~3 reflect the limited number of meteorological stations (12) in and around this region.

Considering possible questions surrounding the reliability of the Kriging interpolation of precipitation, we highlight the annual spatial precipitation calculated from the TerraClimate monthly dataset [49] as auxiliary evidence; the TerraClimate annual estimations show a gradual increase in rainfall from 2005 to 2015 (Figure 5). Regression analysis indicates a significantly positive correlation between precipitation and runoff (R = 0.4259, $\rho$ = 0.047), whereas there is no significant correlation between temperature and runoff (R = 0.0366, $\rho$ = 0.737), suggesting that the increase in runoff has a bigger effect on precipitation than temperature (Figure 4c).

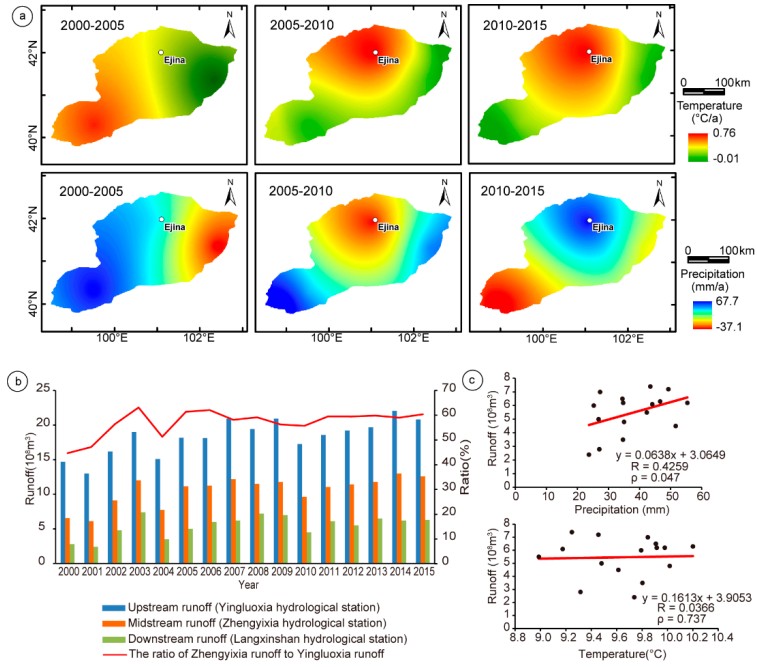

**Figure 4.** Analyses of meteorology and runoff in the HRDB from 2000 to 2015. (**a**) Spatial variations of temperature and precipitation in the HRDB obtained by Kriging interpolation. (**b**) Runoffs of the Heihe River recorded by hydrological stations. (**c**) The correlations between the annual runoff and climate factors (precipitation and temperature) in the HRDB. R: correlation coefficient; $\rho$: *p*-value of *t*-test ($\rho < 0.05$ indicating a significant trend at 0.05 level).

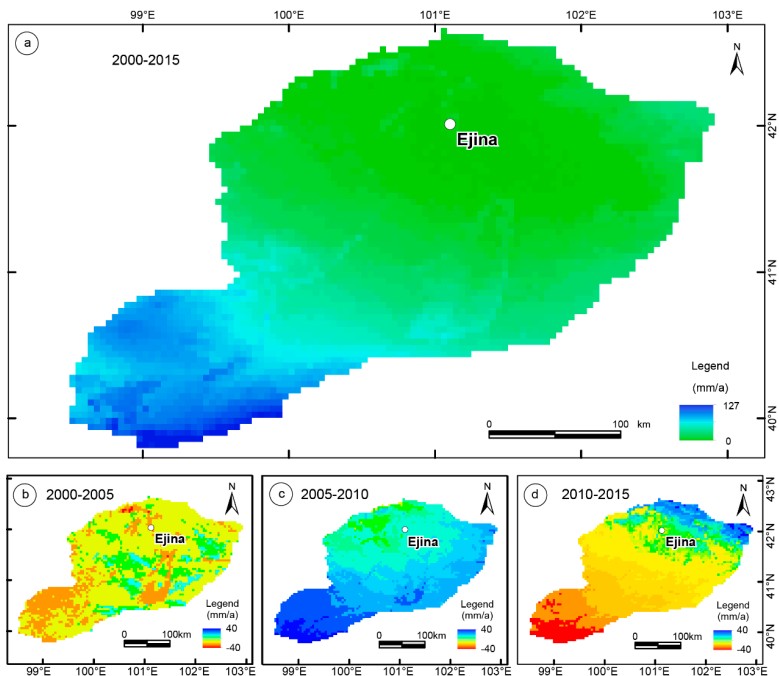

**Figure 5.** The spatial features of annual precipitation calculated from the TerraClimate monthly dataset in the HRDB from 2000 to 2015. (**a**) The average annual precipitation in HRDB from 2000 to 2015. Spatial variations of precipitation in three periods: (**b**) 2000–2005, (**c**) 2005–2010, and (**d**) 2010–2015. The TerraClimate monthly dataset was downloaded from the Climatology Lab (http://www.climatologylab.org/terraclimate.html).

### 3.3. Water Thresholds for Runoff Allocation along the Heihe River Basin

Runoff data from hydrological stations indicated changes in the upstream, midstream, and downstream from 2000 to 2015 (Figure 4b and Table S4). The runoff data from the Zhengyixia station, which recorded the amount of water dispatched from midstream to downstream (see the location of the stations in Figure 1), show an upward trend from $6.6 \times 10^8$ to $12 \times 10^8$ m$^3$ from 2000 to 2003. The average runoff is $10.6 \times 10^8$ m$^3$ from 2000 to 2015. It is worth noting that the ratio between the runoff from the Zhengyixia station and that from the Yingluoxia rose from 44.9% to 62.3% with an average of 57.6%, suggesting a significant increase in the water amounts allocated to the HRDB.

The thresholds of the water balance between the downstream and midstream are determined based on their water demands. Previous researchers have estimated the ET to be in the range of 611–978 mm/a within the Heihe River Basin over the past 15 years [31,55,58,59] (Table 5). Although the precipitation has increased by 26–52 mm/a, it is far below the levels needed to meet the ET consumption in the HRDB. Accordingly, this study indicated an addition of at least ~$5.06 \times 10^8$ m$^3$ of water was needed from runoff to both safeguard ecological resilience and meet socio-economic needs (Figure 6 and Table 5) without serious degradation of the HRDB oases. In addition, Liu and Shen (2017) showed that the midstream area required a minimum flow of ~$6.83 \times 10^8$ m$^3$ from the Heihe River, excluding the discharge from other rivers, precipitation, and the use of groundwater. The average annual flow at the Yingluoxia station of ~$18.3 \times 10^8$ m$^3$ is equal to the total water amount allocated to the midstream and downstream [60,61]. Accordingly, the maximum thresholds of the downstream and midstream were estimated to be ~$11.49 \times 10^8$ m$^3$ and ~$13.26 \times 10^8$ m$^3$, respectively (Figure 6).

**Table 5.** Water demand amounts in the HRDB.

| Year | Average ET [1] (mm/a) | Precipitation (mm/a) | Total Area (km$^2$) | Vegetation Area (km$^2$) | Lake Area (km$^2$) | Industrial Water ($10^8$m$^3$) | Domestic Water ($10^8$m$^3$) | Water Demand ($10^8$m$^3$) |
|------|------|------|------|------|------|------|------|------|
| 2000 | 796.69 | 27.05 | 66,574.9 | 3984.41 | 0.00 | 0.006 | 0.006 | 13.746 |
| 2005 | 611.40 | 26.80 | 66,574.9 | 4380.84 | 38.93 | 0.055 | 0.004 | 9.237 |
| 2010 | 978.2 | 51.61 | 66,574.9 | 3947.32 | 70.13 | 0.113 | 0.005 | 5.056 |
| 2015 | 850 | 46.60 | 66,574.9 | 5757.40 | 70.73 | 0.083 | 0.008 | 18.608 |

[1] The average annual ET is obtained from previous results [31,55,58,59].

If the water flow exceeds the threshold either in downstream or midstream, either may suffer water shortages. The green area in Figure 6a presents the rational water allocation values, within which the downstream ecosystem can be well restored without causing water shortages in midstream. Meanwhile, due to global warming, higher temperatures and more vegetation will increase evaporation [37], and more glacier-snow-melting water will increase the total water amounts. As inferred by the blue area in Figure 6a, the thresholds of water demand in the downstream and midstream may increase if the total runoff is higher than the current average of $18 \times 10^8$ m$^3$. Considering the fluctuation of the thresholds, this study suggests the ratio of water dispatching to the downstream and midstream should remain between 0.4 and 1.7, as an index to determine whether the water allocation is rational.

The verification of the water thresholds is based on the true runoff recorded from the Yingluoxia and Zhengyixia stations from 2000 to 2015. Calculations show that the average annual water allocated to the downstream is $10.6 \times 10^8$ m$^3$, accounting for about $p = 57.6\%$ of the upstream flow, while the midstream water accounts for $q = 42.4\%$ of the upstream flow. Together, these results prove that the ecosystem of the HRDB has been effectively restored because the water allocation ratio, $p/q$, is 1.4—which is between 0.4 and 1.7, consistent with the rational allocation ratio suggested by our model (Figure 6b).

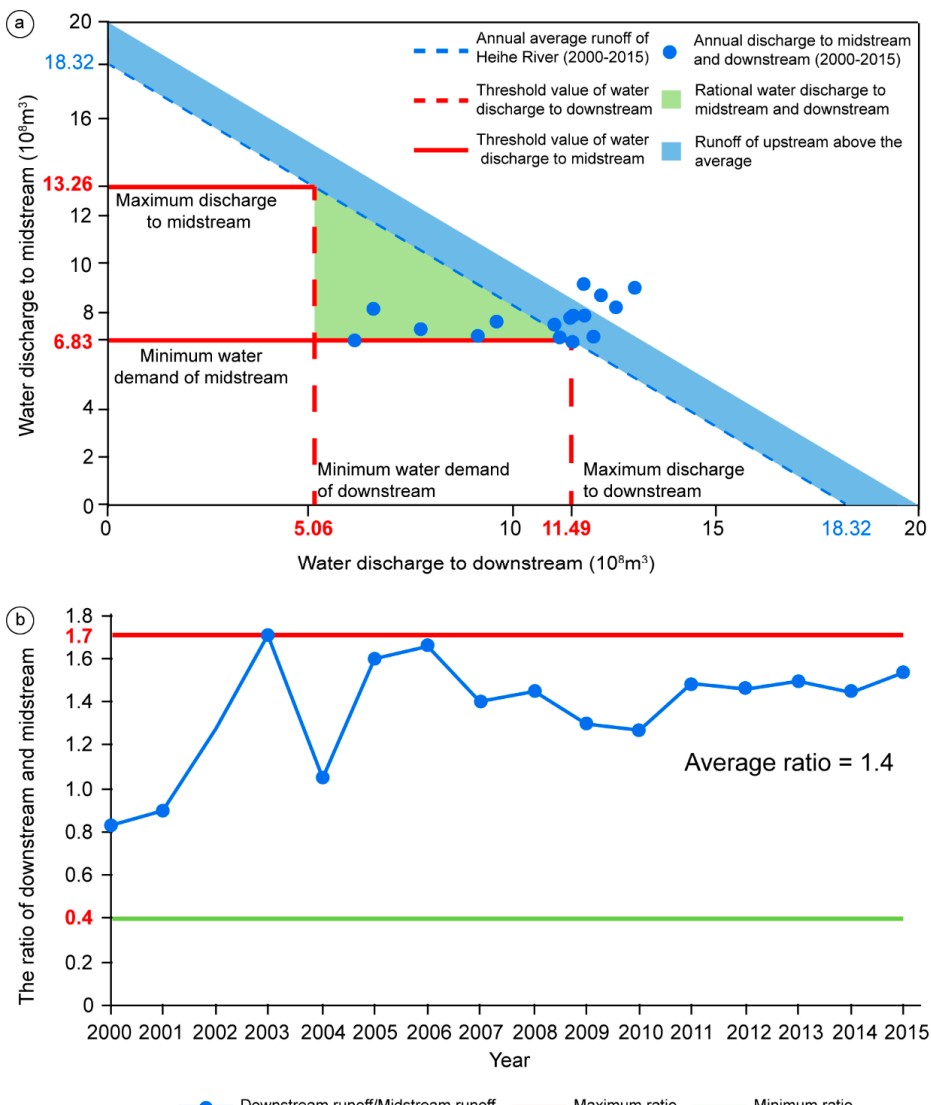

**Figure 6.** Water balance model for the Heihe River basin. (**a**) The rational water thresholds allocated to the downstream and midstream can restore the HRDB's ecological resilience without water shortages occurring in midstream when the runoff values are within the green areas. (**b**) The verification for the ratio of downstream and midstream was based on their water discharge thresholds by using the true runoff value from 2000 to 2015, showing that the ecosystem can be effectively restored when the runoff ratio ranges from 0.4 to 1.7.

## 4. Discussion

Coupled with Landsat satellite data and a decision tree, we assessed the desertification dynamics of five periods over 20 years. Our results highlight a significant reversion of desertification in the HRDB since ecological water dispatching began in 2000. About 26,119 km$^2$, or 20% of the total catchment area, was restored from severe desertification. The areas of vegetation and lake were both recovered in the HRDB. Meanwhile, considering that a ~1 °C increase in temperature can induce potential evapotranspiration (ET) of around 75 mm/a [62], the ET increased by approximately 4.5 mm/a as the temperature increased by 0.06 °C/a in the HRDB from 2000 to 2015. Although the increase in precipitation (+1.22 mm/a) was far from satisfying the water requirements of ecosystem restoration, spatio-temporal monitoring also indicated the positive impact of water dispatching on precipitation in the HRDB from 2000 to 2015, which is helpful for improving the ecological resilience of oases.

Water availability is crucial for oasis development [33,34]. It is necessary to consider the water balance related to the ecological types of different oasis ecosystems for sustainable development within inland river basins [8,63,64]. Field investigations indicate the changes in vegetation type with different amounts of water along the Heihe River basin (Figure 7). In the upstream, the meadow and shrub are the main plants nurtured by glacier-snow-melting water from "Asian water towers". The recovery of the meadow ecosystem plays an important role in the regulation of the hydrological change in the frozen soil of the Tibetan Plateau [65]. In the midstream, the vegetation is characterized as arbor, shrub, and crop. The cultivated lands consumed 80% of the oasis water, which had a negative impact on the meeting of natural water demands [57]. Toward the downstream, the significant restoration of shrub and meadow has been improved, though the vegetation is still sparse. The increase in vegetation also plays a positive role in sand prevention and fixation.

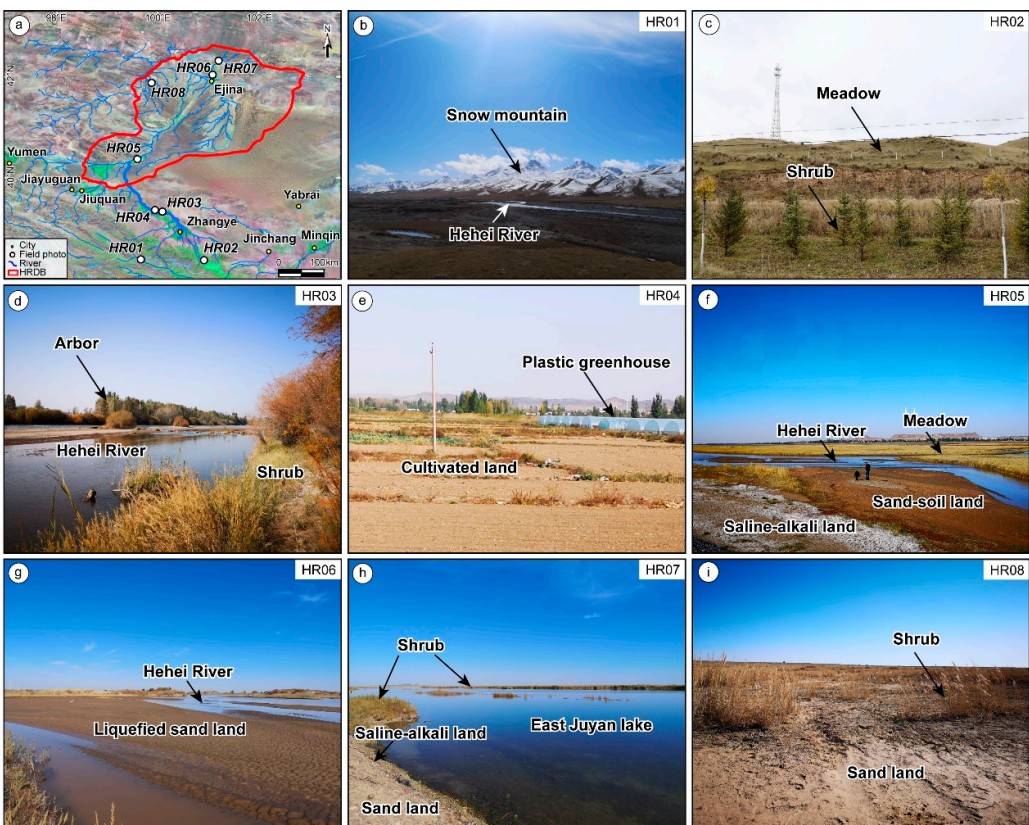

**Figure 7.** Field photographs showing the ecological features along Heihe River. (**a**) Photographs of the field locations are presented. (**b**,**c**) Ecological features in the upstream of the Heihe River. (**d**–**f**) Agricultural patterns in the midstream of the Heihe River. (**g**–**i**) Illustrations of ecological restoration in the HRDB.

There is currently more glacier-snow melting from the AWT discharge to the upstream runoff due to global warming [1,9,66], contributing to the ecosystem's recovery in the HRDB. However, long-term global warming may cause the volumes of glaciers to decline, leading to a potential decrease in runoff and drought in the HRDB [1,66]. Besides, the continuous over-dispatching of water to the HRDB probably results in water shortages in midstream [8,57]. Thus, rational water allocation is critical for maintaining a long-term, sustainable "triple bottom line" and improving the ecological resilience of the oasis. Accordingly, a reasonable water dispatching threshold is needed to balance ecological resilience and socio-economic development along the Heihe River basin. As estimated above, the current maximum threshold for water allocated to the HRDB should not exceed $\sim11 \times 10^8$ m$^3$,

and the ratio of water allocated to the downstream and midstream should remain at 0.4–1.7, which is important for the balance of water consumption in the midstream and downstream.

According to the successful water dispatching along the Heihe River basin, we propose a conceptual model to provide new insights into the water balance allocation in trans-provinces along the Silk Road (Figure 8). As shown in Figure 8, the effective implementation of water dispatching requires the consideration of different water resource problems among the upstream, midstream, and downstream. In the upstream, snow- and glacier-melting from the Qilian Mountain contributed 28.5% of water to the runoff, which is facing the negative impacts from long-term global warming [9]. In the midstream, as a central area of the river basin, its highly efficient regulation and utilization of water are critical for maintaining the "triple bottom line" within the whole drainage region, since the increasing urban, industrial, and agricultural developments consume the greatest amounts of the available water. In the downstream, the river runoff determines the vegetation restoration and wet-land recovery since there has low precipitation and strong evaporation. This study suggests that water allocation between the downstream and midstream accounting for 57.6% and 42.4% of the total upstream flow, respectively, could balance the ecological and socio-economic water demands along the Heihe River basin.

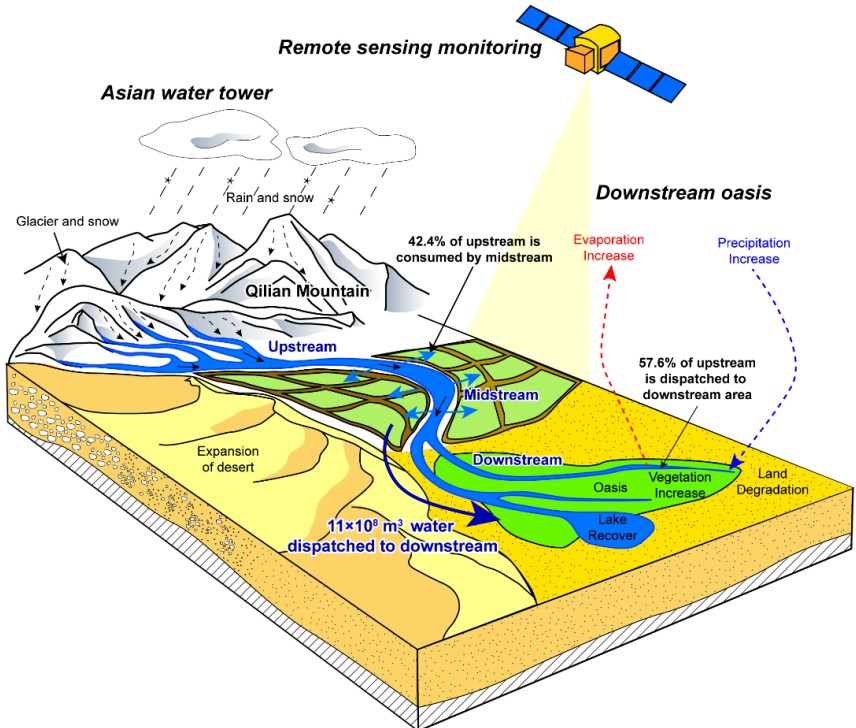

**Figure 8.** A three-dimensional conceptual model of a water balance diagram showing the pattern of the inland river basin. Upstream flow is discharged from the Asian water tower. Water consumption in the midstream and downstream accounts for 42.4% and 57.6% of the upstream flow, respectively. The annual water dispatching threshold is $11 \times 10^8$ m$^3$. The current water allocation ratio (1.4) between the downstream and midstream can well balance the water requirements in oases along the Silk Road.

This experience can provide a useful reference for addressing the ecological problems in oases with similar river patterns. For example, the over-consumption of water in the midstream of Amu Darya and Syr Darya rivers had caused a drought in their terminal lake-Aral Sea basin since the 1960s, where the water volume decreased by about 90% and lake areas had shrunk by 600–870 km$^2$/a [6,63,67]. The lack of an effective agreement among Kazakhstan, Kyrgyzstan, Tajikistan, Turkmenistan, and Uzbekistan on the allocation of water resources from these two river basins has exacerbated the ecological crisis of the Aral Sea basin. Trans-boundary coordination and cooperation among the five Central Asian countries is needed to optimize the water utilization among different areas in the Aral Sea basin [68].

As we have shown here, doing so will require the long-term, accurate assessment of desertification to determine water thresholds for water allocation strategies.

## 5. Conclusions

To summarize, combining Landsat TM/OLI images analysis embedded in a decision tree analysis informed by meteorological, hydrological, and water utilization data provide an effective approach for the long-term monitoring and assessment of ecological dynamics in inland river basins. The MNDWI, MSAVI, and BSI can synthetically reflect the ecological restoration of the Ejina oasis in the HRDB from 1995 to 2015 due to the implementation of the ecological water dispatching project. The assessment results show significant recovery of the degraded lands by ~69%, with a restoration of vegetation coverage and lakes, as well as an increase in precipitation in the HRDB since 2000. Furthermore, we estimate that maximum water amounts allocated to the downstream of ~11 $\times$ 10$^8$ m$^3$, with an allocation threshold between the downstream and midstream of 0.4–1.7, are required for a sustainable "triple bottom line" water balance. Generally, the comprehensive remote-sensing-based monitoring and assessment of ecological dynamics indicate that the ecological resilience of the oases is improved by water dispatching, playing a key role in optimizing the decisions regarding and implementation of water management policies in arid and semi-arid regions.

**Supplementary Materials:** The following are available online at http://www.mdpi.com/2072-4292/12/16/2577/s1, Figure S1: Mann-Kendall inspection of climate change, Table S1: Annual average temperature and precipitation of Heihe River downstream basin from 2000 to 2015, Table S2: Cross validation of Kriging interpolation for precipitation in the HRDB1 from 2000 to 2015, Table S3: Cross validation of Kriging interpolation for temperature in the HRDB1 from 2000 to 2015, Table S4: Runoffs of upstream, midstream and downstream of the Heihe River from 2000 to 2015.

**Author Contributions:** Conceptualization, B.F.; methodology, J.D. and Q.G.; software, J.D. and Q.G.; validation, J.D., Q.G. and P.S.; formal analysis, J.D. and Q.G.; investigation, J.D., Q.G. and P.S.; resources, J.D., Q.G. and P.S.; data curation, J.D., Q.G. and P.S.; writing—original draft preparation, J.D.; writing—review and editing, J.D. and B.F.; visualization, J.D. and B.F.; supervision, B.F.; project administration, B.F.; funding acquisition, B.F. All authors have read and agreed to the published version of the manuscript.

**Funding:** This research was funded by the Strategic Priority Research Program of Chinese Academy of Sciences (XDA 20070202), the National Natural Science Foundation of China (No. 41761144071), the Remote Sensing Geological Survey Of Key Earth Belts (DD20190536) and the West Young Scholars Program, West Light Foundation of Chinese Academy of Sciences (2017–XBQNXZ–A–009, 41372214).

**Acknowledgments:** We sincerely acknowledge the constructive suggestions given by Mike Sandiford from the University of Melbourne that greatly helped to improve the manuscript.

**Conflicts of Interest:** The authors declare no conflict of interest.

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
