# Peer review of "Monitoring and Assessment of the Oasis Ecological Resilience Improved by Rational Water Dispatching Using Multiple Remote Sensing Data: A Case Study of the Heihe River Basin, Silk Road"

_remotesensing, doi:10.3390/rs12162577_

Round 1

Reviewer 1 Report

The paper if of intetrest tp the readers of the journals.

  1. The English need to be improved!
  2. It would be neccessasry to have a section on the statistics used in the paper in the Methdos section
  3. Why were the 3 specific indices selected? What were there advantages?
  4. Why was the the specific water balance equation chosen?
  5. How were th data frm the weather stations (point data) transformed in to spatial for the entire watershed. 

Author Response

Dear Reviewer,

Thank you very much for providing us this unique opportunity and the detailed revision of our manuscript Remotesensing-859249 submitted to Remote Sensing. We really appreciate your constructive suggestion that help us improve the content and many key points of the text, tables and figures, from which we have learned a lot as well.

The outline of major revision is listed below. The details of the revision for replying each suggestion and comment point-by-point are appended followed.

List of major revision

  • Introduction

In the revised manuscript, the introduction has reviewed the literatures about the similar research on remote sensing applied to desertification monitoring and assessment around the world, as well as the explanation of “Asian water tower”.

  • Method

There is a major change in Method section from previous 4 sections to 6 sections. In section 2.3, the reason for selection of 3 indices was explained and the processing of decision tree for desertification assessment was depicted in detail. In new section 2.4, the statistics data, including meteorological data, hydrological data and water utilization data, as well as their utilization were introduced. In new section 2.5, the details of spatial-temporal analyses of climate data were added, including Kriging interpolation, cross validation, Mann-Kendall test, spatial analysis of TerraClimate data and the correlation analysis. In section 2.6, the water balance model equations and variables were depicted in detail one by one separately.

  • Figures

The Figure 2 (previous the Supplementary Figure S1) was inserted into Method section 2.3 to illustrate the decision tree rule and the range values of 3 indices in determining the desertification degree. The Figure 3 was mainly revised, the images of small scale in previous figure were deleted as suggested.

  • Tables

The new Table 1 was added to Method section 2.2, listing the information of Landsat imageries used for desertification assessment in this study.

  • Conclusion

As suggested, the Conclusion has been revised to one paragraph.

  • Supplementary materials

There are two new Supplementary Tables 2 and 3 were added in the revised Supplementary materials, listing the cross validation values of precipitation’s and temperature’s Kriging interpolation results.

Details of revision for replying each suggestion and comment point-by-point.

  1. The English need to be improved!

Re: The English of the revised manuscript has been carefully improved, particularly the gramma and spell. And we have also consulted some native English speaker for some details about language usage.

  1. It would be necessary to have a section on the statistics used in the paper in the Methods section

Re: (Lines 185-198) As suggested, the new section 2.4 introduces the statistics used in revised manuscript, including meteorological data, hydrological data and water utilization data. The acquisition resource and application of these statistics data were depicted in this section in detail.

  1. Why were the 3 specific indices selected? What were there advantages?

Re: (Lines 139-144) The section 2.3 of revised manuscript has explained the limitation of other index, like NDVI, SMA, and RUE documented in previous study. The advantages of the 3 indices were depicted separately, which can cover the shortage of other indices listed above.

  1. Why was the specific water balance equation chosen?

Re: (Lines 223-225, 226-227, 228-229) In new section 2.6 of revised manuscript, the reason for each water balance equation was explained in detail before the each equation.

  1. How were the data from the weather stations (point data) transformed in to spatial for the entire watershed.

Re: (Lines 200-217) As suggested, the new section 2.5 in revised manuscript have depicted the Kriging interpolation of spatial analyses of temperature and precipitation data in detail. We downloaded the monthly data of meteorological station from China Meteorological Administration (http://data.cma.cn/), which was calculated to the annual averaged data in Arcgis 10.2. The Kriging interpolation and cross validation were conducted in spatial analyst module of Arcgis 10.2.

Thank you again for reviewing the revised manuscript. If you have any questions, please don’t hesitate to let us know. Please see the attachment for the word version if needed.

Reviewer 2 Report

Monitoring and assessment of the oasis ecological resilience improved by rational water dispatching in the Heihe River basin, Silk Road

This paper provides a model for monitoring and assessment of spatio-temporal ecological dynamics for Silk Road's oasis ecosystem. It combines Landsat satellite data from 1995 to 2015 and decision tree algorithm for determination of water dispatching and rational water dispatching thresholds.

This manuscript needs a major revision before it is considered for potential publication in this journal. My major concerns are listed as follows:

- Lines 69-74: the Authors mention that there is a lack of similar investigations in the HRDB. Perhaps the Authors should connect with similar research which is mentioned in Discussion (Lines 318 – 319), nevertheless Introduction needs to be strengthened with similar research along with quantitative measures

- I suggest to insert a Table in Section 2.2. for Landsat imagery, along with ancillary information about time of acquisition, path/row, spatial resolution etc.

- Section 2.3. – Have the authors considered to include additional Landsat indices in order to obtain more robust results, e.g. reference [33] quantifies vegetation change in semiarid environments by using Spectral Mixture Analysis (SMA) and Normalized Difference Vegetation Index (NDVI) applied to Landsat TM data, but none of the aforementioned index was not included in this research

- After Equations (1) – (3) please describe the spectral bands used in order to calculate Landsat index

- Last paragraph in Section 2.3. needs to be improved. It is not clear from where come the ground references and from (photo interpretation ? government database ? ) and how you split between train and validation. Perhaps the authors should include Fig. S1 into a main body of the paper

- Tables 1 and 2 are very hardly readable since the second digit of the Area column transfers into a new line. Aforementioned Tables need to be adjusted

- Line 157/185/270 : spatio-temporal?

- Figure S2: abbreviations UF and UB are not described

- Figure 3: Right part of the Figure is cut-off, and therefore the whole content of the Figure is not visible

- Reference Liu and Shen (2017) are mentioned in Line 231 and referenced afterwards [45] in Line 280?

- Figure 6 indicate field investigations of various ecological and agricultural features in HRDB. Since the results were computed from 1995 to 2015, was there a consideration of including an additional Landsat imagery from 2020, or using it for independent validation of suggested results?

I recommend to reconsider after major revision, but there is a lot a revision to process.

Author Response

Dear Reviewer,

Thank you very much for providing us this unique opportunity and the detailed revision of our manuscript Remotesensing-859249 submitted to Remote Sensing. We really appreciate your constructive suggestion that help us improve the content and many key points of the text, tables and figures, from which we have learned a lot as well.

The outline of major revision is listed below. The details of the point-by-point revision for replying each suggestion and comment are appended followed.

List of major revision

  • Introduction

In the revised manuscript, the introduction has reviewed the literatures about the similar research on remote sensing applied to desertification monitoring and assessment around the world, as well as the explanation of “Asian water tower”.

  • Method

There is a major change in Method section from previous 4 sections to 6 sections. In section 2.3, the reason for selection of 3 indices was explained and the processing of decision tree for desertification assessment was depicted in detail. In new section 2.4, the statistics data, including meteorological data, hydrological data and water utilization data, as well as their utilization were introduced. In new section 2.5, the details of spatial-temporal analyses of climate data were added, including Kriging interpolation, cross validation, Mann-Kendall test, spatial analysis of TerraClimate data and the correlation analysis. In section 2.6, the water balance model equations and variables were depicted in detail one by one separately.

  • Figures

The Figure 2 (previous the Supplementary Figure S1) was inserted into Method section 2.3 to illustrate the decision tree rule and the range values of 3 indices in determining the desertification degree. The Figure 3 was mainly revised, the images of small scale in previous figure were deleted as suggested.

  • Tables

The new Table 1 was added to Method section 2.2, describing the information of Landsat imageries used for desertification assessment in this study.

  • Conclusion

As suggested, the Conclusion has been revised to one paragraph.

  • Supplementary materials

There are two new Supplementary Tables 2 and 3 were added in the revised Supplementary materials, presenting the cross validation values of precipitation’s and temperature’s Kriging interpolation results.

Details of point-by-point revision for replying each suggestion and comment.

  1. Lines 69-74: The Authors mention that there is a lack of similar investigations in the HRDB. Perhaps the Authors should connect with similar research which is mentioned in Discussion (Lines 318 – 319), nevertheless Introduction needs to be strengthened with similar research along with quantitative measures

Re: (Lines 77-89) As emphasized by the constructive suggestion, the Introduction of revised manuscript reviews the similar research of remote sensing (RS) in desertification around the world, focusing on the types of RS data applied to monitor and assess land degradation, the desertification related impacts and drive factors.

  1. I suggest to insert a Table in Section 2.2. for Landsat imagery, along with ancillary information about time of acquisition, path/row, spatial resolution etc.

Re: (Line 133) As suggested, a new Table 1 has been inserted in Section 2.2, which lists the basic information of Landsat imagery used in this study, including the type, dates of acquisition, path/row, spatial resolution and data download source etc.

  1. Section 2.3. – Have the authors considered to include additional Landsat indices in order to obtain more robust results, e.g. reference [33] quantifies vegetation change in semiarid environments by using Spectral Mixture Analysis (SMA) and Normalized Difference Vegetation Index (NDVI) applied to Landsat TM data, but none of the aforementioned index was not included in this research

Re: (Lines 139-144) The section 2.3 of revised manuscript has explained the limitation of NDVI and SMA in assessing arid HRDB. Previous works have indicated the NDVI is seriously affected by soil background and rainfall. In addition, sparse vegetation probably reduces the accuracy of SMA. Thus, considering in HRDB, where vegetation is sparse, annual rainfall is less than 50 mm per year and runoff has a greater impact than rainfall, we choose MNDWI, MSAVI and BSI to comprehensively reflect the water, vegetation and soil status in HRDB to better conduct desertification assessment.

  1. After Equations (1) – (3) please describe the spectral bands used in order to calculate Landsat index

Re: (Lines 160-161) As suggested, the spectral bands for Landsat-5 TM and Landsat-8 OLI used to calculate monitoring indicators have been explained one by one after Equations (1) – (3).

  1. Last paragraph in Section 2.3. needs to be improved. It is not clear from where come the ground references and from (photo interpretation? government database?) and how you split between train and validation. Perhaps the authors should include Fig. S1 into a main body of the paper

Re: (Lines 162-178) In Section 2.3 of revised manuscript, we explained that the training samples and validation points have been selected refer to GlobeLand30 data (http://www.globallandcover.com/GLC30Download/) and ground features of Google Map. A total of 1,500 training samples for each index, including 300 training samples of each desertification grade, was selected to obtain decision tree rule for classification of desertification. And a total of 1,500 validation points for Landsat imageries of each period, including 300 validation points of each desertification grade, was used to verify the desertification results of each period. Previous Fig. S1 has been inserted in the revised manuscript as Figure 2.

  1. Tables 1 and 2 are very hardly readable since the second digit of the Area column transfers into a new line. Aforementioned Tables need to be adjusted

Re: (Lines 263-264) Tables 1 and 2 have been changed as Tables 2 and 3. The format of them have been readjusted to fit the page.

  1. Line 157/185/270: spatio-temporal?

Re: Previously, I have no idea which is correct, “spatial-temporal” or “spatio-temporal”, since I found these two forms both appear in previous publication and they have the same translation in translator. Anyway, all the “spatial-temporal” have been uniformly revised as “spatio-temporal” in this revised manuscript, because I think the “spatial” is used alone, where as “spatio” is used in phrase, such as “social” and “socio-economic”, though I am still not sure which is better and more appropriate. I would appreciate if you could provide a good suggestion.

  1. Figure S2: abbreviations UF and UB are not described

Re: Previous Figure S2 has been changed to Figure S1. The explanation of UF and UB has been added in the caption of Figure S1.

  1. Figure 3: Right part of the Figure is cut-off, and therefore the whole content of the Figure is not visible

Re: (Line 294) Previous Figure 3 has been changed to Figure 4. The embedding format of Figure 4 has been adjusted to fit.

  1. Reference Liu and Shen (2017) are mentioned in Line 231 and referenced afterwards [45] in Line 280?

Re: (Line 244) As suggested, in the revised manuscript version, the Reference Liu and Shen (2017) has firstly been referenced afterwards [57] in new section 2.6

  1. Figure 6 indicate field investigations of various ecological and agricultural features in HRDB. Since the results were computed from 1995 to 2015, was there a consideration of including an additional Landsat imagery from 2020, or using it for independent validation of suggested results?

Re: Thanks for your interesting and constructive suggestion. Sincerely, we indeed considered an additional validation analysis for 2019 or 2020. However, we find the amount of water dispatching almost reaches or even exceed the threshold we estimated in this study, and the desertification restoration probably has a turning point due to negative impact from the long-term over-water dispatching after 2015. Thus, we are preparing the next-step systematic study on desertification status and associated impacts of the potential over-water dispatching from midstream to downstream from 2015 to 2020 in HRDB. We would really appreciate if you’d like to provide any constructive suggestion for our further research.

Thank you again for reviewing the revised manuscript. If you have any questions, please don’t hesitate to let us know. Please see the attachment for the word version if needed.

Reviewer 3 Report

I thank the authors to work on this interesting project. i found the paper well written but I missed specific questions in the introduction. I suggest the authors to put the research questions as well as do some lit review. Please follow my suggestions below.

  1. The introduction has scopes for improvement. I encourage the authors to do a literature review on the use of remote sensing (RS) in environmental assessment (focusing on desertification and its associated impacts). The authors should discuss what types of studies (RS or not) have been conducted on their study area as well as how RS has been used to assess environmental/ecological impacts (in any part of the world). Also, define what is 'Asian water tower'.
  2. Line 136-140: Write each variable name separately. 
  3. In the method section, for each index, tell us how to interpret. For example, the range of values for MNDWI and what do those values mean. 
  4. Present how you classify the desertification degree in the main manuscript, not as a supplementary.
  5. Line 156: The category name "non" and "slight" do not sound good. Suggest rename.
  6. In fig 2, remove the small scale map- it is repetitive. 
  7. The result and the following parts look alright.
  8. Make the conclusion one paragraph.

Author Response

Dear Reviewer,

Thank you very much for providing us this unique opportunity and the detailed revision of our manuscript Remotesensing-859249 submitted to Remote Sensing. We really appreciate your constructive suggestion that help us improve the content and many key points of the text, tables and figures, from which we have learned a lot as well.

The outline of major revision is listed below. The details of the revision for replying each suggestion and comment point-by-point are appended followed.

List of major revision

  • Introduction

In the revised manuscript, the introduction has reviewed the literatures about the similar research on remote sensing applied to desertification monitoring and assessment around the world, as well as the explanation of “Asian water tower”.

  • Method

There is a major change in Method section from previous 4 sections to 6 sections. In section 2.3, the reason for selection of 3 indices was explained and the processing of decision tree for desertification assessment was depicted in detail. In new section 2.4, the statistics data, including meteorological data, hydrological data and water utilization data, as well as their utilization were introduced. In new section 2.5, the details of spatial-temporal analyses of climate data were added, including Kriging interpolation, cross validation, Mann-Kendall test, spatial analysis of TerraClimate data and the correlation analysis. In section 2.6, the water balance model equations and variables were depicted in detail one by one separately.

  • Figures

The Figure 2 (previous the Supplementary Figure S1) was inserted into Method section 2.3 to illustrate the decision tree rule and the range values of 3 indices in determining the desertification degree. The Figure 3 was mainly revised, the images of small scale in previous figure were deleted as suggested.

  • Tables

The new Table 1 was added to Method section 2.2, listing the information of Landsat imageries used for desertification assessment in this study.

  • Conclusion

As suggested, the Conclusion has been revised to one paragraph.

  • Supplementary materials

There are two new Supplementary Tables 2 and 3 were added in the revised Supplementary materials, listing the cross validation values of precipitation’s and temperature’s Kriging interpolation results.

Details of revision for replying each suggestion and comment point-by-point.

  1. The introduction has scopes for improvement. I encourage the authors to do a literature review on the use of remote sensing (RS) in environmental assessment (focusing on desertification and its associated impacts). The authors should discuss what types of studies (RS or not) have been conducted on their study area as well as how RS has been used to assess environmental/ecological impacts (in any part of the world). Also, define what is 'Asian water tower'.

Re: (Lines 77-89) As emphasized by the constructive suggestion, in the Introduction of revised manuscript reviews the literature of the application of remote sensing (RS) in desertification around the world, focusing on the types of RS, the desertification related impacts and drive factors.

(Lines 36-38) Additionally, the term of “Asian water tower” has been explained in detail in the first paragraph of introduction.

  1. Line 136-140: Write each variable name separately.

Re: (Lines 235-241) As suggested, each variable name has been written separately one by one.

  1. In the method section, for each index, tell us how to interpret. For example, the range of values for MNDWI and what do those values mean.

Re: (Lines 150-166) In Section 2.3 of revised manuscript, we indicate each index extracted according to spectral calculation equations (1)-(3) for processed-Landsat imagery of each period in ENVI.

(Line 178) Previous Fig. S1 has been inserted in the revised manuscript as Figure 2, which shows the range of values for MNDWI, MSAVI and BSI with respect to five desertification degree, respectively.

  1. Present how you classify the desertification degree in the main manuscript, not as a supplementary.

Re: (Lines 162-176) The extraction of each index, selection of training samples and validation points, the details of the decision tree rule for classification of the desertification degree has been presented by Figure 2 and section 2.3.

  1. Line 156: The category name “non” and “slight” do not sound good. Suggest rename.

Re: (Line 164) As suggested, the category name “non” has been changed to “Non-degraded”, “slight” has been changed to “Low”.

  1. In fig 2, remove the small scale map- it is repetitive.

Re: (Line 270) In the revised manuscript, the previous Fig.2 has been revised as Figure 3, where previous maps with scale of 10 km were deleted. Only the map with scale of 100 km reserved to show the change in vegetation and lake of the study area, within which the red rectangle is employed to emphasize the change in Ejina oasis.

  1. The result and the following parts look alright.

Re: Thanks for this comment.

  1. Make the conclusion one paragraph.

Re: (Lines 448-461) As suggested, the conclusion has been revised as one paragraph.

Thank you again for reviewing the revised manuscript. If you have any questions, please don’t hesitate to let us know. Please see the attachment for the word version if needed.

Round 2

Reviewer 1 Report

Dear authors

Thank you for addressing most concerns.

Still English needs improvements.

Some additional comments

Figure 1. Should in the study Area section and not the Introduction.              

Maybe separate Figure 1. The map should be in Study areas and b and c should be in Figure 3a and b for the results.

L278 Missing “the” before “precipitation”

L282. Missing a “is” before “probably.

The example above indicates the need to improve English

L287-290. Even for the precipitation the R is low. Also is this R or R2?

L357-358. Did you take field measurements? What type? Was this mentioned in the methods?

Figure 7b. These shrubs are not natural but planted!

Author Response

Dear Reviewer,

Thank you very much for providing our manuscript Remotesensing-859249 with this unique opportunity and further comments. We really appreciate your constructive suggestion that is helpful to improve the quality of our manuscript.

The details of the point-by-point revision for replying each suggestion and comment are appended below.

  1. Still English needs improvements.

Re: Sure, the English of manuscript has been polished by an English native speaker, Prof. Mike Sandiford, a professor of Earth Science from the University of Melbourne, Australia. I hope this could work. If it needs any further improvement, please let us know.

Some additional comments

  1. Figure 1. Should in the study Area section and not the Introduction.

Re: (Line 120) As suggested, the Figure 1 has been placed in the study area of Section 2.1 in the latest revised manuscript.

  1. Maybe separate Figure 1. The map should be in Study areas and b and c should be in Figure 3a and b for the results.

Re: (Lines 120 and 280) As suggested, the Figure 1 was separated and was placed in Study areas of Section 2.1. The previous Figures 1b and 1c were revised to be Figures 3a and 3b placed in Section 3.1.

  1. L278 Missing “the” before “precipitation”

Re: (Line 317) As suggested, the “the” has been added before “precipitation” in Section 3.2 of revised manuscript.

  1. Missing a “is” before “probably.

Re: (Lines 299-333) This section was revised in new manuscript. 

  1. L287-290. Even for the precipitation the R is low. Also is this R or R2?

Re: This is R. Although the R for the precipitation and temperature is both low, we suggested that the runoff has relatively bigger impact on the improvement of precipitation than that of temperature.

  1. L357-358. Did you take field measurements? What type? Was this mentioned in the methods?

Re: We did not take field measurements. We mainly identify the ground objects along the Heihe River and we want to display different ecological features in the upstream, midstream and downstream, respectively, in Figure 7. This allows us to suggest that the ecological difference in different watershed areas along a river should be taken into account for a rational water dispatching project.

     8. Figure 7b. These shrubs are not natural but planted!

Re: Yes, most shrubs along the Heihe River are planted for combating desertification. And the types of shrubs differ with the variation of ecological environment along the Heihe River as shown in Figure 7. The rational water dispatching contributes to the growth of both natural and planted vegetation for improving desertification restoration.

Thank you again for your helpful comments and suggestions.

Please find attachment for word version if needed.

Reviewer 2 Report

The authors provide significant answers and updates from v1 to v2.

My suggestion to the authors would be to keep “spatio-temporal” in manuscript. 

I have just two more technical notes: 

  • at the end of the manuscript the authors describe what is included in Supplementary materials, so I do not think that it is necessary to indicate in the body of the manuscript "Supplementary Table S4" (e.g., Line 299, 278), I think that "Table S4" would be just enough
  • Table 2 - if the authors agree, OA can be rounded to 2 decimal places in order to adjust with other statistical indicators

Author Response

Dear Reviewer,

Thank you very much for providing our manuscript Remotesensing-859249 with this unique opportunity and further comments. We really appreciate your constructive suggestion that is helpful to improve the quality of our manuscript.

The details of the point-by-point revision for replying each suggestion and comment are appended below.

  1. My suggestion to the authors would be to keep “spatio-temporal” in manuscript.

Re: Thank you very much for your suggestion!

I have just two more technical notes:  

  1. at the end of the manuscript the authors describe what is included in Supplementary materials, so I do not think that it is necessary to indicate in the body of the manuscript "Supplementary Table S4" (e.g., Line 299, 278), I think that "Table S4" would be just enough

Re: As suggested, all the “Supplementary” indicated in the body of the latest revised manuscript are deleted.

  1. Table 2 - if the authors agree, OA can be rounded to 2 decimal places in order to adjust with other statistical indicators

Re: As suggested, in the latest revised manuscript, the OA numbers are changed to 2 decimal places in Table 2.

Thank you again for reviewing the revised manuscript.

Please find attachment for word version if needed.
